# Neuronutraceuticals Combating Neuroinflammaging: Molecular Insights and Translational Challenges—A Systematic Review

**DOI:** 10.3390/nu14153029

**Published:** 2022-07-23

**Authors:** Shakta Mani Satyam, Laxminarayana Kurady Bairy

**Affiliations:** Department of Pharmacology, RAK College of Medical Sciences, RAK Medical and Health Sciences University, Ras Al Khaimah 11172, United Arab Emirates; satyam@rakmhsu.ac.ae

**Keywords:** neuroinflammation, aging, nutraceuticals, neurodegenerative diseases, neuroprotection

## Abstract

Neuropathologies, such as neuroinflammaging, have arisen as a serious concern for preserving the quality of life due to the global increase in neurodegenerative illnesses. Nowadays, neuronutraceuticals have gained remarkable attention. It is necessary to investigate the bioavailability, off-target effects, and mechanism of action of neuronutraceuticals. To comprehend the comprehensive impact on brain health, well-designed randomized controlled trials testing combinations of neuronutraceuticals are also necessary. Although there is a translational gap between basic and clinical research, the present knowledge of the molecular perspectives of neuroinflammaging and neuronutraceuticals may be able to slow down brain aging and to enhance cognitive performance. The present review also highlights the key emergent issues, such as regulatory and scientific concerns of neuronutraceuticals, including bioavailability, formulation, blood–brain permeability, safety, and efficacy.

## 1. Introduction

Neuroinflammation is hypothesized to develop from prolonged immune system activation in the central nervous system (CNS) in response to numerous insults. This brain immune stimulation leads to resident cell activation and invasion of the CNS by circulating immune cells. Additionally, pro-inflammatory mediators that lead to neurodegenerative disorders are produced and secreted. The malfunctioning of the central nervous system (CNS) may be a cause of a variety of human medical conditions associated with both cognitive impairment and varying degrees of neuroinflammation. A growing body of research indicates that the central nervous system’s (CNS) homeostasis depends on a two-way exchange of information between the brain and the immune system [1].

The process of aging can be thought of as being multifactorial because it results from the interaction of genetic, environmental, and lifestyle variables. The notion that cerebral aging is a complex multifactorial process has supplanted earlier monofactorial theories that linked the aging phenomenon to a single cause. With increasing age, there is a significantly increased risk of neurological problems, particularly neurodegenerative disorders. The production of toxins that result in the death or malfunctioning of neurons in neurodegenerative illnesses may be significantly influenced by age-dependent heightened neuroinflammatory processes. According to a meta-analysis of data from around the world, the prevalence was 40.51 per 100,000 people aged 40 to 49, 106.67 per 100,000 people aged 50 to 59, 428.48 per 100,000 people aged 60 to 69, 1086.54 per 100,000 people aged 70 to 79, and 1902.98 per 100,000 people aged 80 and more [2].

The term “neuroinflammaging” refers to the up-regulation of the inflammatory response and a low level of chronic neuroinflammation that accompany neuronal aging [3,4,5,6]. Neuroinflammaging is characterized by five conditions: low-grade, controlled, asymptomatic, chronic, and neuroinflammation [4]. Neuroinflammaging is characterized by the up-regulation of a number of stress responses. The interplay of genetics, environment, and lifestyle choices results in multiple and complicated mechanisms that cause neuroinflammaging and the accompanying neurodegenerative disorders. A better understanding of the connections between neuroinflammaging and neurodegenerative disorders could therefore result in new therapeutic strategies. Given that pro-inflammatory cytokine levels are elevated with aging, numerous therapies may be successful in bringing these levels down. The neuroinflammaging process may also produce reactive oxygen species (ROS), which cause oxidative damage and increase cytokine production. This vicious cycle results in a chronic proinflammatory state where tissue damage and repair mechanisms happen at the same time. Furthermore, oxidative stress can influence other transcription factors, such as nuclear factor-kB (NF-kB) and activator protein-1, and also induce apoptosis [3]. Some of the most important aspects of immunosenescence are the accumulation of memory cells, the growth of megaclones, the restriction of the repertoire of T lymphocytes, and the increase in autoimmune symptoms [5]. The body’s capacity to adapt correctly to a variety of oxidative stressors and inflammation appears to have a significant role in promoting human longevity and in preventing/delaying the main age-related illnesses. In human brains, aging is linked to abnormal inflammatory responses [7,8]. Specifically basal levels of pro-inflammatory cytokines are enhanced with aging [9], whereas anti-inflammatory mediators are lowered [10]. The complement (C) pathway, toll-like receptor (TLR) signaling, and inflammasome activation are additional elements involved in innate immune responses that are similarly elevated as the brain ages [7,11,12]. In reality, as we become older, our brains produce different levels of pro-and anti-inflammatory cytokines.

Numerous nutraceuticals and traditional medicines frequently exhibit pleiotropic effects, which include positive impacts on brain health as well as immunomodulatory and anti-inflammatory benefits. Broad ranges of naturally derived substances with nutritional and medicinal qualities to improve brain health are referred as “neuro-nutraceuticals”, [13]. Neuronutraceuticals often have one or two substances, such as vitamins and fatty acids. There is a paucity of clinical data regarding their overall safety, efficacy, and possible drug/food interactions. Furthermore, it has been noticed that while healthy people may not benefit from brain supplements, sick people may. As a result, their use needs to be given additional thought [14,15,16].

The fundamental idea behind modern drug research is that medicine should have only one site of action and few or no side effects that could jeopardize its selectivity. Though there are numerous instances of clinically licensed medications acting on multiple targets in the literature, numerous attempts have been undertaken to create medications that have dual-purpose activities. The adage “the whole is more than the sum of its parts” may be worth reevaluating in this situation; perhaps it is the numerous beneficial effects of a single “effective” nutraceutical rather than the combined effects of several nutraceuticals found in a functional food that support health and treat disease conditions [17]. In the current context of molecular and cellular neuroscience, a number of mechanisms, such as oxidative stress, excitotoxicity and their downstream effects, are involved in neurodegenerative illnesses and brain injury. However, inflammation (recruitment of micro-/macroglial signaling), decreased cerebral blood flow, and alterations in the blood–brain barrier are also associated with neuronal injury. As a result, it would not be speculative to suggest that one naturally occurring and bioavailable chemical in a mixture of potentially bioactive molecules may interact with many targets to have several ameliorative effects on CNS. Due to their well-balanced composition of the majority of nutrients beneficial to the brain, including flavonoids, carotenoids, saponins, polyunsaturated fatty acids, marine natural products, probiotics, vitamins, etc., dietary herbal formulations have attracted the most interest as neuronutraceuticals. Starting with the fact that inflammation is a crucial factor in aging and chronic degenerative diseases, and that several nutraceuticals can disrupt and control cellular processes, this review aims to explore the key molecular perspectives of neuroinflammaging processes and neuronutraceuticals in combating the same. In addition, translational challenges associated with neuronutraceuticals are highlighted in the current review.

## 2. Methodology

The PRISMA guidelines were followed in conducting this systematic review (an evidence-based minimum set of items for reporting in systematic reviews and meta-analyses for systematic reviews).

### 2.1. Search Strategy

The literature was searched on PubMed, Scopus, EMBASE, and the Cochrane Central Register of Controlled Trials (CENTRAL) from inception to 31 December 2021, during the first 6 months of the year 2022. All the keywords for the literature search were used as indexing terms, such as MeSH (if available) and text words. The following essential words and their synonyms were used and truncated occasionally: Neuroinflammaging, Neuroinflammation, Neuronal aging, Neuronutraceuticals, and Nutraceuticals. The literature search was done using the following search equations in the titles and the abstracts: “Neuronutraceuticals OR Neuroinflammaging”, “Nutraceuticals AND Neuroinflammation”, “Nutraceuticals AND Neuronal aging”. We scrutinized the articles based on the title first and then the abstract (Figure 1). Later, we gathered information from the selected relevant articles by reading the full text and highlighting the facts relevant to our research question. Grey literature was looked through and references were cross-checked to look for relevant papers that had not come up in earlier searches. There was no language restriction on the literature search.

### 2.2. Study Eligibility Criteria

#### 2.2.1. Types of Studies

##### Included

We incorporated pre-clinical studies (mostly rodent disease models), case-control studies, prospective and retrospective cohort studies, and randomized controlled trials (RCTs). The interventions of interest were directly contrasted in RCTs and non-randomized studies. We considered both prospective and retrospective research for non-randomized investigations.

##### Excluded

Due to the inferior quality of the evidence, we did not include descriptive studies, cross-sectional studies, conference presentations, case series, or case reports. We also excluded studies due to (1) insufficient data such as short communications, letters, brief reports, and comments, (2) duplicate articles, (3) lack of access to study data even by contacting the first author or corresponding author, and (4) clinical studies without control or placebo groups.

## 3. Literature Search Results

The results of the literature search have been discussed below under two sections: Section 3.1 Neuroinflammaging from a molecular perspective (Figure 2), Section 3.2 Neuronutraceuticals and neuroinflammaging.

### 3.1. Neuroinflammaging from a Molecular Perspective

#### 3.1.1. Microglia

Microglia are abundantly distributed throughout the brain and are the main innate immune cells as well as the early responses to pathogenic insults [18,19]. Depending on their activation level, microglia can either be pro-inflammatory or neuroprotective in the CNS. Pro-inflammatory cytokines, which are byproducts of infections or injured cells, cause resting microglia to express pro-inflammatory molecules such as nitric oxide (NO), proteases, IL-1, TNF-α, and IL-6, which are harmful in neurodegenerative disorders [20,21]. Contrarily, IL-4, IL-10, IL-13, and transforming growth factor (TGF)- activate neuroprotective microglia, which causes the production of numerous substances including FIZZ1, Chitinase-3-Like-3 (Chi3l3), Arginase 1, Ym1, CD206, insulin-like growth factor (IGF-1), and Frizzled class receptor 1 (Fzd1) [20,21,22,23]. These microglial components could contribute to tissue repair and neuroprotection.

Major Histocompatibility Complex II (MHC II) and CD11b are two glial activation markers that are up-regulated in microglia with normal aging. Young animals’ microglia express MHCII at incredibly low levels during basal conditions, giving a clear baseline for identifying alterations in microglia immunophenotype related to aging [24]. Age-related increases in microglia numbers or rises in per microglial cell expression could both lead to increased MHCII [23,24]. Although there is research, that supports the notion that increasing per microglial cell expression leads to sensitization [25].

Glial activation, inflammatory mediators, and brain shrinkage have all been shown in studies to significantly increase with age [26,27]. Age-associated changes in the expression of genes related to cellular stress, inflammation, and trophic support have been reported [28]. These alterations imply that as neurons age, they face greater difficulties while also receiving less support.

The cause of increasing inflammation with aging is still unknown. Genetic studies reveal that DNA plays a substantial role in this process and that DNA bases are particularly vulnerable to oxidative stress damage, which causes considerable inflammatory alterations [29].

The extent to which aging changes microglia’s responsiveness or their capacity to cause neuronal death is also unknown. Despite morphological and phenotypic changes that imply microglial activation, it has been proposed that microglia may actually become dysfunctional and enter a senescent state with age [27]. Such a condition may lead to decreased neurotrophic factor secretion and a downregulation of phagocytic activity in microglia. This process may cause neuronal death and ineffective removal of harmful protein clumps in neurodegenerative diseases since it is linked to increased secretion of inflammatory mediators [26].

#### 3.1.2. Astrocytes

The majority of glial cells in the brain are astrocytes [30]. Similar to microglia, astrocytes are pro-inflammatory, and they have immunoregulatory (neuroprotective) capacity. Pro-inflammatory reactive astrocytes are known to upregulate a number of genes, including complement cascade genes and generate pro-inflammatory chemicals, such as IL-1 and TNF-α, which are known to have detrimental consequences [20,21]. In contrast, several neurotrophic factors and thrombospondins are upregulated by the neuroprotective reactive astrocytes [21]. In a neuroprotective manner, anti-inflammatory cytokines, including IL-4, IL-13, and IL-10, can activate astrocytes. These activated astrocytes can then produce IL-4, IL-10, and TGF-β [31]. Pro-inflammatory microglia can generate inflammatory mediators, such as IL-1, IL-6, TNF-α, and C1q, which can activate pro-inflammatory astrocytes and trigger a subsequent inflammatory response [32,33].

A large number of other cytokines, sphingolipids, and neurotrophins can all cause detrimental astrocytic signaling pathways [30]. During neuro-inflammatory conditions, astrocytes activate the receptors for IL-17 and tropomyosin receptor kinase B. NF-kB activator 1 (Act1) may be attracted to IL-17 receptors and pro-inflammatory cytokines may be produced as a result [34]. The JAK-STAT3 pathway could mediate reactive astrocytes’ neuroprotective properties. Uncertainty exists about the chemical mechanism underlying the development of neuroprotective reactive astrocytes [35]. There may be additional polarization states than only the pro-inflammatory or the neuroprotective [21].

#### 3.1.3. Toll-like Receptors (TLRs)

The innate immune system is significantly influenced by a class of proteins known as TLRs. They are single-pass membrane-spanning receptors that identify pathogen-derived compounds with structural conservation that are typically expressed on sentinel cells, such as macrophages and dendritic cells. Multiple toll-like receptors (TLRs) are expressed more often in the aged brain, which may cause glia and neurons to become hypersensitive and amplify possible damage [36]. The stimulation of TLRs results in the activation of NF-kB and the subsequent transcriptional activation of a large number of proinflammatory genes, including those encoding adhesion molecules, immunological receptors, cyclooxygenase 2, inducible nitric oxide synthase enzymes, cytokines, and complement proteins [37].

#### 3.1.4. Mitogen-Activated Protein Kinases (MAPKs)

The MAPK family of serine/threonine protein kinases mediates the majority of cellular reactions to cytokines and outside stressors, which is also essential for controlling the synthesis of inflammatory mediators. The stress-activated protein kinase/c-Jun N terminal kinase (JNK), the extracellular signal-regulated kinases (ERKs) and the p38 MAPK are three subfamilies that are divided into MAPK transduction cascades depending on how much the sequences are similar [38]. In response to various stimuli, these kinases are connected to pathways that lead to both survival and death to control cellular functions, such as cell division, proliferation, survival, and differentiation. However, depending on the type and function of a specific neuron, activation of ERKs can either be neuroprotective or can cause cell death [39].

JNK may be an important regulator of the inflammatory and apoptotic pathways that are activated in the course of neuroinflammaging and neurodegenerative disorders, according to a significant body of literature [40,41,42]. In fact, JNK has a role in the inflammatory responses of astrocytes and primary glial cells. JNK activity has also been linked to reports that it controls neuronal survival by preserving mitochondrial homeostasis [1]. Last but not least, mounting evidence points to the specific roles that p38 plays in aging and neuroinflammation [43].

#### 3.1.5. Sirtuins

The significance of class III histone deacetylases, also known as sirtuins, in neurodegenerative processes has been demonstrated by a wealth of evidence [44,45,46]. The primary described sirtuin (SIRT1) molecule controls the formation of ROS via regulating immunological responses through NF-kB signaling [47]. Studies have reported the antioxidant and anti-inflammatory actions of SIRT1 [48,49]. In particular, SIRT1 has a strong NF-kB signaling inhibitory effect, which reduces inflammation [50,51]. One of the studies reported that microglial SIRT1 deficiency causes cognitive deterioration in healthy aging [11]. The scientists hypothesized that an increased expression of IL-1β is linked to memory impairments and accompanying cognitive decline, which could be caused by epigenetic changes brought on by SIRT1 deficiency (aging-induced) in microglia. Activating SIRT1 and other sirtuins may also protect neurons in experimental models of neurodegenerative diseases, according to a growing body of research [52]. A prospective therapy alternative for neurological diseases, such as Alzheimer’s, Parkinson’s, and Huntington’s disease as well as for the prevention and the advancement of neuroinflammaging, is now thought to include SIRT1 and the other sirtuins [53].

#### 3.1.6. Nuclear Factor (erythroid-derived)-like 2 (Nrf2)

Nrf2 is another important molecule connected to neuroinflammaging. Some researchers have suggested that Nrf2 has an antagonistic effect on the NF- kB pathway, which is regarded as a hallmark of neuroinflammation [54,55]. Nrf2 regulates the antioxidants necessary for cells to protect themselves from various electrophiles and oxidants [56,57]. Nrf2 has been linked to the development of several beneficial proteins, including brain-derived neurotrophic factor (BDNF) [58], the anti-apoptotic B-cell lymphoma 2 (BCL-2) [59], the anti-inflammatory IL-10, and the mitochondrial transcription co-factors Nrf1 and peroxisome proliferator-activated receptor [60]. Although the relationship between Nrf2 and NF-kB is not fully understood, the discovery of NF-kB binding sites in the Nrf2 gene’s promoter region implies that there may be an interaction between these two inflammatory process regulators [61]. According to a study, as compared to their wild-type littermates’ hippocampi, Nrf2 deletion mice were more vulnerable to the inflammation caused by lipopolysaccharide (LPS), which was shown to increase microglial cells and the inflammatory markers iNOS, IL-6 and TNF-α [62]. The increase in peripheral IGF-1 that flows into the brain after exercise may also activate Nrf2 [63,64]. Rojo et al. recently demonstrated that Nrf2-deficient animals displayed increased astrogliosis and microgliosis [65]. COX-2, iNOS, IL-6, and TNF-α, inflammation indicators linked to traditional microglial activation were likewise elevated, while FIZZ-1 and IL-4, anti-inflammatory signals linked to alternative microglial activation, were lowered. These findings were verified in microglial cultures, illustrating even more clearly how Nrf2 regulates the harmony between conventional and unconventional microglial activation [65].

BBB permeability rises in old animals, possibly allowing monocyte infiltration that releases ROS produced by mitochondria [66,67]. CD11bC and CD45 cell counts, which are indicative of infiltrating monocytes, have been found to rise with age in the brains of elderly rats, supporting this notion [66]. Similar to this, the hippocampal region expresses chemotactic molecules, such as interferon-inducible protein 10 (IIP10) and monocyte chemotactic protein-1, at higher levels than normal (MCP-1) [66,68].

#### 3.1.7. Oxidative Stress

There are studies indicating that reactive oxygen species/reactive nitrogen species (ROS/RNS) can stimulate inflammation via the activation of inflammasomes, and the production of IL-1 and IL-18 cytokines subsequently trigger inflammatory responses [47,69,70]. Oxidative stress and inflammation interact with one another in a complex manner. The production of ROS and RNS accelerates the synthesis of a variety of chemokines and pro-inflammatory cytokines, such as IL-1, IL-6, and tumor necrosis factor (TNF-α), during brain aging. These inflammatory mediators cause microglia and astrocytes to become highly ROS/RNS-producing cells. Studies have revealed that neuroinflammatory reactions could be viewed as the result of ongoing oxidative stress [71,72].

### 3.2. Neuronutraceuticals and Neuroinflammaging

The use of neuronutraceuticals is intended to improve brain health. Many phytochemicals share a shared lineage with conserved biological processes, such as the similarities in most pathways for the production and breakdown of macro biomolecules. This common ancestry is assumed to be the cause of these substances’ potential impacts [73]. In fact, all eukaryotes include a number of compounds that function as neurochemicals in the mammalian central nervous system (CNS) [74]. The use of nutritional supplements rich in polyphenols as a modulator of age-related cognitive decline is justified by the age-related increase in oxidative stress and low-grade inflammation [75,76]. Phytochemicals are often thought to have health benefits because of their natural anti-inflammatory and antioxidant capabilities [39]. Phytochemicals can be harmful in large concentrations yet have positive or stimulating effects on animal cells in low doses [77]. Hormonal phytochemicals have been shown to promote cellular resistance to injury and illness. These compounds include resveratrol, sulforaphane, curcumin, catechins, allicin, and hypericin [78]. We have enlightened below an overview of some of the neuronutraceuticals having the potential to mitigate neuroinflammaging (Table 1).

#### 3.2.1. Curcuma Longa

Curcuma longa also known as turmeric is a perennial shrub that is widely distributed throughout Southeast Asia. This herb’s underground stems (rhizome) are a crucial part of toothpaste, food additives, and spices used in cooking. Nitric oxide release, NF-kB signaling, astrocytes and microglia (reactive), cytokine production (IL-23, IL-1, IL-6, TNF-α), and PPAR transcriptional activity were all inhibited by curcumin to reduce neuroinflammation [79,80,81]. Additionally, numerous studies demonstrated that curcumin interacts with NF-kB and this connection regulates T-cell-mediated immunity in a protective manner [82]. The creation of mitochondrial ROS, activation of the transcription factors Nrf2 and NF-kB, elevation of the protein kinase Mitogen-Activated Protein Kinase (MAPK) p38, and suppression of phosphatase activity were all associated with curcumin-induced HO-1 overexpression in rats and human cells [83,84]. Curcumin inhibits the synthesis of pro-inflammatory mediators via modifying the histone acetylation of transcription factors and the methylation pattern of genes implicated in the inflammatory response [85]. Curcumin’s neuroprotective effects also entail the regulation of SIRT1. Curcumin’s neuroprotective impact is thought to be mediated by SIRT1 signaling activation. Prophylactic administration of curcumin (50 mg/kg) reduced inflammation, apoptosis, and mitochondrial dysfunction in the ischemic brain among rats [86].

#### 3.2.2. Anthocyanins

These are a group of flavonoids that includes water-soluble colored pigments. Berries that are red, blue, or purple are one of the main sources of dietary anthocyanins [87]. Anthocyanins (100 mg/kg) have been found to inhibit the release of pro-inflammatory mediators and shield cellular components from oxidative damage brought on by demyelination [88]. Additionally, prolonged use of a blackberry anthocyanin extract (25 mg/kg) may have favorable effects on synaptogenesis and synaptic plasticity while reducing the negative effects of neuroinflammation in high-fat-fed mice [89]. Through a variety of processes, such as the control of Nrf2 and the suppression of NF-kB pathways, anthocyanins shield CNS from pro-oxidant and inflammatory damage [90]. In a rodent model of the peripheral nervous system, improved myelination following red wine treatment is thought to be caused by SIRTs activation, particularly SIRT1 [91].

#### 3.2.3. Flavanols

Flavanols such as catechin and epicatechin were discovered in cocoa in higher concentrations than in other plant-based diets. The modulation of several pathways is thought to be how flavanols affect neuroinflammaging. In both pre- and post-treatment, the injection of an epicatechin (30 mg/kg) dose dependently protects against transitory ischemia-induced brain injury through activation of the Nrf2/HO1 pathway in rodents [92]. Catechin inhibits the synthesis of pro-inflammatory mediators and reduces NF-kB activation through modulation of the ERK and p38 MAPK pathways in LPS-induced BV-2 microglial cells [93]. Epigallocatechin gallate (EGCG), one of the components of green tea, up-regulates haem oxygenase-1 (HO-1) expression via activating the Nrf2-ARE pathway in endothelial cells, imparting resistance against Hydrogen peroxide (H_2_O_2_) induced cell death and indicating a hermetic method of action [94]. It has been shown that EGCG specifically guards against oxidative stress in cultured rat cerebellar granule neurons [95].

#### 3.2.4. Resveratrol

Resveratrol is usually present in grapes, red wine, mulberries and peanuts. It is well known that resveratrol reduces peripheral inflammation. Resveratrol impacts numerous pro- and anti-inflammatory variables because it crosses the blood–brain barrier [96,97]. Resveratrol’s wide anti-inflammatory properties and potential advantages during neuroinflammation are shown by the fact that it successfully suppressed proinflammatory cytokines in activated macrophage and microglial cell lines at quantities detected in plasma. Studies have shown that resveratrol has neuroprotective benefits, particularly against beta-amyloid-induced oxidative cell death and against dopaminergic neuronal injuries [98,99]. Resveratrol pretreatment reduced the transiently induced activation of NF-kB by Aβ, indicating a critical role for the NF-kB inflammatory pathway in the deposition of A and a potential therapeutic benefit of resveratrol in mediating neuroprotection [98]. Resveratrol has been demonstrated to influence SIRT1 activity in vitro depending on the kind of deacetylation substrate [100]. Resveratrol does not affect the NF-kB proteins’ ability to bind to DNA, but it did prevent reporter gene transcription and the TNF-α-induced translocation of the NF-kB p65 subunit. Resveratrol also inhibits the activation of JNK and its upstream MAPK, which may provide insight into how it suppresses AP-1 [101]. Overall, the findings of this study indicate that resveratrol protects neurons from hypoxia-induced neurotoxicity via inhibiting inflammation in microglia. These effects were at least partially achieved by inhibiting the NF-kB, ERK, and JNK/MAPK signaling pathways from being activated [102].

#### 3.2.5. Oleuropein and Hydroxytyrosol

These are ingredients found in virgin and extra-virgin olive oils (VOO and EVOO), which are derived from the fruits of the Olea europea. Tyrosol, hydroxytyrosol, and oleuropein, which are phenolic substances found in olive oil, can reduce the impact of the chronic inflammatory milieu on glioblastoma by regulating TNF-α, COX-2, JNK, ERK, and NF-kB [103]. In the cortex of the brains of db/db mice, hydroxytyrosol also enhances neuronal survival and mitochondrial function, and it lowers oxidative stress. After 8 weeks of administration at doses of 10 and 50 mg/kg, the energy-sensing protein network activated SIRT1, and Nrf2 are known to modulate mitochondrial function and oxidative stress responses [104].

#### 3.2.6. Bacopa Monniera

Also known as Brahmi or BM, this herb belongs to the Scrophulariaceae family. It is a creeping perennial herb. It is widely distributed throughout the tropical United States and East Asia, including India, in wetlands and on the shores of lakes and rivers. Ayurvedic, Unani, Siddha, and homeopathic traditional medical literature all make special reference to BM for its energizing, nootropic, and mental health-promoting properties. These preliminary studies highlighted the anti-inflammatory (100 mg/kg BW) and antioxidant (dose range: 40–250 mg/kg BW, duration: 1–4 weeks) potential of BM extracts [105,106,107]. In addition to reversing cell cycle arrest, BM also reduced intra-neuronal protein aggregation and lipofuscin accumulation, and it prevented microglia from secreting pro-inflammatory cytokines (IL-6 and TNF-α) in aging and dementia models [107,108].

#### 3.2.7. Withania Somnifera

This herb is also referred to as Indian ginseng, poison gooseberry, and ashwagandha (WS, Family: Solanaceae). This plant grows best in the dry climates of South and Central Asia, Africa, India, Pakistan, Bangladesh, Sri Lanka, Afghanistan, Egypt, the Middle East, and North America. This herb is grown commercially in numerous nations, including India, and it is listed among the selected therapeutic plants in World Health Organization monographs [109]. Animals treated with WS exhibited reduced levels of reactive gliosis, inflammatory cytokines such as TNF-α, IL-1β, and IL-6, and expression of nitro-oxidative stress enzymes. Additionally, an investigation of the NF-kB, P38, and JNK MAPK pathways revealed their contribution to the inhibition of neuroinflammation [110].

#### 3.2.8. Ferulic Acid (FA)

This is frequently present in a variety of foods, including tomatoes, sweet corn, and rice [111]. FA lowers inflammatory mediator levels (prostaglandin E2 and TNF-α), as well as iNOS expression and function [112,113]. By preventing microglial activation in vivo, long-term treatment of FA effectively guards against Aβ toxicity [114].

#### 3.2.9. Sulforaphane (SFN)

Sulforaphane, a phytochemical found in high concentrations in cruciferous vegetables such as broccoli, has been shown to activate the Nrf2-ARE stress response pathway in mouse brains and microvasculature, reducing brain damage in a traumatic brain injury model [115]. Sulforaphane has been found to protect dopaminergic neurons from mitochondrial toxins and oxidative stress in cultured neurons [116,117,118].

#### 3.2.10. Polyunsaturated Fatty Acids (Eicosapentaenoic and Docosahexaenoic Acids)

These reduce the inflammatory response of activated microglial cells, and these anti-inflammatory effects seem to be helpful in preventing age-related memory impairment [119]. The favorable effects of polyunsaturated fatty acids in ischemic stroke and traumatic brain damage models may be attributed to AMPK and Nrf2, which limit microglial activation, similar to resveratrol [120].

#### 3.2.11. Sallyl Cysteine

The most prevalent molecule in aged garlic extracts has a variety of neuroprotective properties that are probably mediated by antioxidant actions, presumably via Nrf2, but more recent data imply furthermore a diversity of anti-inflammatory effects [121]. Another component of garlic-Allicin, efficiently inhibited neuronal injury by acting on sphingosine kinase-2 against idle cerebral artery blockage in rats [122].

#### 3.2.12. Gut Microbiota

The targeted dietary and probiotic uptake appear to have a good effect on the treatment of some age-related illnesses, and they also provide a viable therapeutic alternative for the aging process [21]. Nowadays, numerous studies are being conducted to explore the regulation of the gut–brain axis with an influence on healthy aging and mental health as a result of a growing body of evidence showing a favorable association between the health of the gut microbiota and brain health. The gut–brain axis is a two-way chemical communication system that transmits data from the intestine to the brain via soluble chemical signals as well as sympathetic and parasympathetic nervous system inputs. The gut microbiota is a third component that has a significant impact on the health of both systems. Aging is associated with behavioral changes, anxiety, and cognitive impairment in addition to synaptic deterioration, oxidative stress, and neuroinflammation. The gut–brain axis is recognized as a crucial, accessible target for fostering healthy brain aging [123,124].

#### 3.2.13. Marine Natural Compounds

These simultaneously act as multiple-target modulators of intestinal and supra-intestinal illnesses. Marine natural products have been reported to modulate the inflammatory mediators, apoptosis, and oxidative stress in the gut, including NF-kB, TNF-α, ILs, COX-2, and TLRs [125]. Additionally, these substances control several important gut-related pathways, such as PI3K/Akt/mTOR, MAMPs, BDNF, and ERK/CREB/MAPK [125]. Citreohybridonol was obtained from the marine-derived fungal strain Toxicocladosporium species [126]. According to their research, citreohybridonol exerts antineuroinflammatory effects by reducing the number of proinflammatory mediators and cytokines produced by activated BV2 cells, including TNF-α, IL-1, IL-12, IL-6, iNOS, and COX-2. Citreohybridonol and related terpenoid derivatives are the targets of this action in order to create anti-inflammatory treatments for illnesses where neuroinflammation has been observed [126].

## 4. Translational Challenges

Any nutraceutical’s translation to humans often faces two significant obstacles: scientific and regulatory. No universally accepted definition exists for the group of products known in various nations as dietary supplements, natural health products (NHPs), complementary medicines, or food supplements. For example, a product that is considered a dietary supplement and is regulated as a food in the USA may be considered a food supplement, a therapeutic good (complementary medicine), a therapeutic good (prescription medication), or even a restricted substance in another country. When nations such as China or India are taken into consideration, where traditional medicine or phytomedicine using unprocessed botanicals already has an established regulatory framework, the situation becomes even more problematic. Numerous regulatory frameworks are evolving, which only makes matters more confusing. The scientific issues and the regulatory frameworks that have developed to address nutraceuticals differ significantly between nations.

Regarding the classifications of human bioactives needs and the implications for dietary supplements, scientists frequently disagree. They disagree about the need for specific non-nutrient bioactives in certain demographic subgroups as well as the potential negative impacts on health from their use. Over a century has passed since the discovery of inborn defects in nutrient metabolism, which can be corrected by giving the deficient nutrient. However, it is unclear whether a paradigm such as this, which is based on single gene abnormalities, is helpful to treat multigenic complicated disorders. It is unknown whether a sizable population of people with prevalent diseases and ailments, such as type 2 diabetes and depression, have specific genetic requirements that necessitate dietary supplements or medical foods [127]. Nutritional implications result from the identification of genetic variations and the development of widely accessible, reasonably priced genetic tests. They have contributed to the emergence of “precision nutrition” or “personalized” eating programs [128], as well as the growth of boutique “personalized” dietary supplements that are purportedly based on a person’s genetic profile. We still do not know how effective these supplements are at preventing chronic degenerative illness.

The effectiveness and the application of analytical procedures play a role in some of the scientific difficulties associated with all of the issues mentioned above. Several unique bioactive compounds found in dietary supplements lack analytical methodologies and reference standards. Whether there should be only one officially recognized method of analysis is still up for debate. Any analytical method that has been correctly calibrated to a recognized reference standard should be adequate. It is the user’s responsibility to demonstrate that the method’s affirmative requirements are met leading to accurate and precise results. Dietary supplements may not respond well to certain techniques that are appropriate for foods.

The safety of dietary supplements largely depends on the dose, aside from problems with product quality. High doses of certain nutrients have a higher propensity than others to produce difficulties, while the amount at which problems arise is disputed. For example, certain dialysis patients who receive extremely high amounts of calcium and vitamin D active on a regular basis may exceed the tolerable upper level (UL) and experience negative health effects, such as calcification of the soft tissues [129]. People with normal renal function may experience negative consequences from extremely high vitamin D dosages [130]. There is not much evidence that the typical dosages and types of these nutrients cause health issues [131]. In countries with programs to fortify their food supplies compared to others, the likelihood of excessive nutrient intake via dietary supplements is higher, so they must also be taken into consideration [132,133,134,135]. It is frequently difficult to establish safe intake amounts of non-nutrient bioactives in supplements due to a lack of dose–response evidence [136,137]. Even after accounting for concurrent use with acetaminophen and alcohol as well as consumption while fasting, some dietary supplements with non-target herbs added purposefully (such as germander as an adulterant for skullcap) or accidentally (such as black cohosh, kava extract, green tea, and others) have been connected to various types of liver damage [138].

**Table 1 nutrients-14-03029-t001:** Summary of the recent update on neuronutraceuticals and neuroinflammaging.

Author/Year	Type of Study	Inference
Liddelow et al., 2017 [28]	Pre-clinical	Gut microbiota modulates age-related illnesses.
Oksanen et al., 2019 [42]	Pre-clinical	Anti-inflammatory cytokines such as IL-4, IL-13, and IL-10 can activate astrocytes in a neuroprotective manner, and these activated astrocytes can then generate IL-4, IL-10, and TGF- β.
Liddelow et al., 2017 [44]	Pre-clinical	Pro-inflammatory microglia can activate pro-inflammatory astrocytes by generating inflammatory mediators and trigger a subsequent inflammatory response
Kumar et al., 2021 [117]	Pre-clinical	Bacopa monniera extract exerts anti-inflammatory (100 mg/kg) and antioxidant (40–250 mg/kg) effects.
Nemetchek et al., 2017 [119]	Pre-clinical	Bacopa monniera decreases intra-neuronal protein aggregation and lipofuscin accumulation and prevents microglia from secreting pro-inflammatory cytokines (IL-6 and TNF-α) in aging and dementia models.
Gupta et al., 2018 [122]	Pre-clinical	Withania somnifera reduces neuroinflammation by decreasing the levels of reactive gliosis, inflammatory cytokines such as TNF-α, IL-1β, and IL-6, and expression of nitro-oxidative stress enzymes and modulating JNK MAPK pathways.
Vaiserman et al., 2017 [135]	Pre-clinical	Gut microbiota has the therapeutic potential for microbiome-targeted interventions in anti-aging medicines.
Fakhri et al., 2021 [137]	Pre-clinical	Marine natural products modulate the inflammatory mediators, apoptosis, and oxidative stress in the gut, including NF-kB, TNF-α, ILs, COX-2, and TLRs, and regulate important gut-related pathways.

Favorable pharmacokinetic profiling of neuronutraceuticals in terms of poor gastrointestinal absorption leading to low bioavailability, poor crossing through the blood–brain barrier, metabolism, and excretion is a major concern. Nutraceuticals must be bioavailable in order to be effective, but the legislation of some nations does not recommend in vitro testing of supplements for disintegration and dissolution, and some products already available in the market do not pass such tests. Researchers and regulators are both concerned about this since it has an adverse effect on studies that look at the effectiveness of dietary supplements. It is possible to examine the dissolution and the disintegration of medications in vitro, and dietary supplement items can be used with these techniques.

Exploration of untoward adverse effects of neuronutraceuticals, especially those related to CNS, such as sedation, tremors, convulsion, psychiatric issues, and interactions with food or other concomitant drugs, is necessary. The issues involved in evaluating efficacy that is pertinent to the testing of all pharmaceuticals include study designs, significance testing, appropriate endpoints, effect sizes, acceptable biomarkers of impact, and the distinctions between clinical and statistical significance. Regulators in some countries require a change in health outcomes or in validated alternative biochemical markers of effect on the causal pathway to a health or performance outcome. Others agree that changes in intermediate biochemical markers, which may or may not be used in place of health outcomes, are acceptable. These concerns have been brought up because it seems that certain countries’ supplements have scant or no evidence of their efficacy. A review of 63 randomized, placebo-controlled clinical trials of dietary supplements in Western adults found that in 45 of them no benefits were observed, 10 showed a trend toward harm, and 2 showed a trend toward benefit, while 4 reported actual injury and 2 reported both harms and benefits. However, only omega-3 fatty acids and vitamin D demonstrated significant enough advantages and no adverse effects to imply potential efficacy [139]. Concerns have been expressed about this hotly discussed topic based on the type of dietary supplement utilized, particularly herbal treatments, the standard of the research included in the review, and other factors, such as the product quality of the supplement being reviewed that must be taken into account [140].

For the safety of nutraceuticals, better chains of custody and product characterization are required than those now in place, especially for those engaging global markets. Effectiveness, or whether the product’s health promotion claims are accurate and not deceptive, is also crucial. Clinical research with clearly defined goods, rigorous experimental designs, and repeatable investigations are necessary for proving efficacy.

Neuronutraceuticals use is common among those who believe it will enhance cognitive function [141]. Another high-risk group includes people who use a combination of prescription, over-the-counter, and dietary supplements, and they require treatments to reduce the possibility of adverse outcomes [142,143]. To advance supplement science, international scientific collaborations are required. Regardless of the type of health product, a successful regulatory framework depends on high-quality science. The scientific data that regulators require must be provided by evaluations of the efficacy, safety, and quality of nutrients and other bioactives [144]. In order to recognize problems and to create solutions, it is essential that scientists and regulators collaborate and share knowledge. While it is necessary to address regulatory obstacles at the national level, it is also important to recognize the global impact of national regulatory decisions on supplements. As Margaret Chan, MD, the former director-general of the World Health Organization stressed, “appropriate regulatory control of nutraceuticals is highly complex and demands that scientists and regulators work together” [145].

## 5. Conclusions

The development of neuroinflammaging may depend on the equilibrium between pro-inflammatory and neuroprotective glial cells. It is necessary to determine the roles played by microglia and astrocytes at particular illness stages in particular patients. There are several theories that relate neuroinflammaging to neurodegenerative disorders, but the processes underlying these connections are still not fully understood.

The evaluation of neuronutraceuticals in short- or long-term nutritional intervention trials depends critically on a better understanding of their mechanisms of action as modulators of cell signaling pathways involved in neuroinflammaging. Numerous neuronutraceuticals are already commercialized for use in treating or preventing various disorders, including age-related cognitive impairment. The majority of these items lack credible scientific support and have not yet received FDA or EFSA approval.

The amounts of phytochemicals, for instance, that pass through the blood–brain barrier and reach the bloodstream are unknown. Bioavailability is still emphasized as a primary concern in human intervention research, despite the fact that delivery technologies, such as nanoparticles, may provide an effective strategy for medication administration into the CNS. Another significant factor contributing to variation and contradictory outcomes is the quality of the chemicals. Understanding whether dietary phytochemicals have adequate effects on particular epigenetic pathways in particular genes or sets of genes is another challenge. In fact, significant changes in epigenetic profiles are related to brain aging, and numerous preclinical investigations have shown that bioactive phytochemicals are crucial in the regulation of total epigenetic modifications (histone modifications, DNA methylations, and microRNA).

The therapeutic application of neuronutraceuticals is constrained by a lack of knowledge on topics, such as their complex metabolic destiny and the role of the gut microbiota in the bioconversion of phytochemicals. Future research focusing on these concerns is required. It is necessary to pay more attention to side effects and possible interactions in order to prevent negative medical results. Before starting or recommending a regimen incorporating these medications, users and doctors should both study the most recent literature. It is important for healthcare professionals to be aware that a significant portion of the general populace uses neuronutraceuticals. Therefore, in order to deliver the best possible medical care, they should ask patients about their supplement use.

## Figures and Tables

**Figure 1 nutrients-14-03029-f001:**
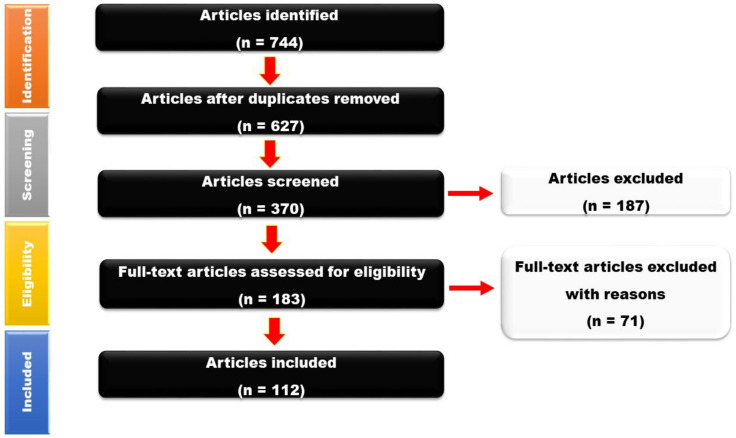
Flowchart showing the process of article selection.

**Figure 2 nutrients-14-03029-f002:**
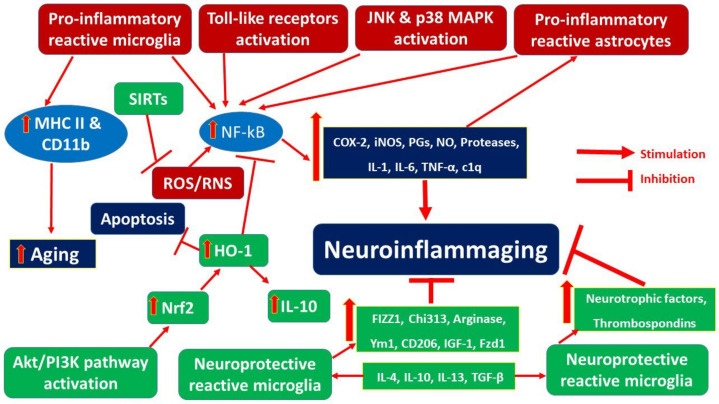
Molecular perspective of neuroinflammaging.

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
