# Peer review of "Neuronutraceuticals Combating Neuroinflammaging: Molecular Insights and Translational Challenges—A Systematic Review"

_nutrients, 2022, doi:10.3390/nu14153029_

Round 1

Reviewer 1 Report

The manuscript from  Shakta Mani Satyam and Laxminarayana Kurady Bairy is a systematic review on the molecular aspects of neuroinflammaging and the use of nutraceuticals. The review satisfies most items in PRISMA 2000 checklist. The authors have properly described the search strategy and the inclusion/exclusion criteria. The authors then describe the molecular mechanisms involved in neuroinflammaging followed by a list of nutraceuticals that have been tested. This is followed by a brief discussion on the translational challenges. Overall, the review seems to be comprehensive and well-written.

The one comment that I have is to identify the manuscript as a systematic review in the title. This is the first item in PRISMA checklist.

Reviewer 2 Report

Dear authors,

You provide a friendly-reading manuscript related to moleculars perspectives of neuroinflammaging processes and neuronutraceuticals in combating aging and chronic degenerative diseases.

However, there are some concerns with regard to the design of this systematic review. All these concerns are referred to the metodhology section and its subsections and to the beginnig of the section Literature Search.

Methodology

1.       You mentioned that the PRISMA guidelines were followed in conducting this systematic review. That’s good, but you should be more explicit in relation to the search equation: was the same equation for all databases?, what search equation was defined?, were necessary the use of MeSH descriptors? Did you need the construction of a PICO question to identify the target population, or is it clear this target population? Please identify the target population in this section.

2.       On the other hand, is this systematic review registered in the PROSPERO database? Please, provide the ID of this study in this database.

3.       It is also relevant to inform about when this search was conducted.

4.       In the inclussion criteria it is necessary to justify why you do not limit the search to a recent period of years (5 year it is usual). In my opinion this limitation is needed to avoid references that even go back more than 15 years and could be interpreted as obsolotes. Here is also important to describe how the selection of articles was carried out, did you select all of the result provided by the search equation, or did you read the title, abstract, full text, reverse search, etc.? Was the level of evidence assessed according to OCEBM?

Literature Search

This section is very well-written from a friendly-reading point of view, but it should include some elements at the beginning.

1.       First, the title, in my opinion, should be ‘Results’.

2.       A Figure related to the selection process for reviewed articles would be very informative. Here you could include the number of identified articles after the search process (Identificaction), then you could describe the screening process with the number of excluded papers after the reading of the abstract, the title an abstract, etc. (Screening), and finally this picture also could include the number of selected documents (Included).

3.       Here it would be very informative for the reader to include a table with a list of the manuscript included in this study. This table could include 7 columns (author/year (country); Desing; Sample; Aims; Support and Technique; Results; Level of evidence/Grade of Recommendation). This is a problem in this moment because you do not limit the range of years and you have too many manuscripts. If my previous recomendation related to the previous number of years limitation in the search equation is considered, then this table could be included.
